# Improved Mechanical Properties of Alumina Ceramics Using Plasma-Assisted Milling Technique

**DOI:** 10.3390/ma16031128

**Published:** 2023-01-28

**Authors:** Shaopeng Tang, Lvping Fu, Huazhi Gu, Ao Huang, Shuang Yang, Renxiang Lv

**Affiliations:** 1The State Key Laboratory of Refractories and Metallurgy, Wuhan University of Science and Technology, Wuhan 430081, China; 2Jinan Ludong Refractories Co., Ltd., Jinan 250109, China

**Keywords:** alumina, mechanical properties, dielectric barrier discharge plasma-assisted milling, sintering

## Abstract

In order to improve the mechanical properties of alumina ceramics, dielectric barrier discharge plasma-assisted milling (DBDPM) was employed to activate alumina powder. The effect of the plasma-assisted milling technique on the grinding behavior of alumina powder, as well as the microstructure and properties of fabricated alumina ceramic, was investigated in detail. Attributed to the great thermal stress induced via plasma heating, DBDPM showed significantly higher grinding efficiency than the common vibratory milling technique. Moreover, the lattice distortion of alumina grains occurred with the application of plasma, leading to an improved sintering activity of the produced alumina powders. Therefore, compared with the common vibratory milling technique, the fabricated alumina ceramics exhibited smaller grain sizes and improved mechanical properties when using alumina powder produced via the DBDPM method as the starting material.

## 1. Introduction

Due to a series of excellent characteristics, such as high hardness, good high-temperature resistance, outstanding oxidation resistance, satisfactory corrosion resistance, high electrical insulation and low dielectric loss, alumina ceramics play an important role in the machinery, chemical, electronics, medicine, aviation and national defense industries [1,2]. However, alumina ceramics produced via commercial process routes typically possess large grain sizes and low mechanical properties, which limit their application range [2,3,4,5,6,7,8].

In order to improve the performance of alumina ceramics [9], plenty of studies have been carried out. For example, extra fine alumina powder was prepared via high-energy ball grinding, and some second phases were added to form composite ceramics [10]. High-energy ball milling is an important technology for the preparation of advanced powder materials [11]. As a powder is ground during ball milling, the specific surface area of the powder increases, and a large number of crystal defects are generated. Therefore, the reactivity of the powder is enhanced after ball milling [12]. This mechanical activation effect of high-energy ball milling is extremely favorable for reducing the subsequent reaction conditions of a powder, and its activation effect and subsequent reaction mode have been extensively studied [13,14]. However, the efficiency of the high-energy ball milling method is low, which leads to long ball milling times and pollution caused by ball milling. Meanwhile, it is difficult to achieve accurate control of material structures and the synthesis of some compounds [14].

In recent years, in order to improve the efficiency of ball milling, it has become an important development direction to superimpose the external physical energy field and the mechanical energy of traditional ball milling. Frequently used external energy fields include ultrasonic [15], magnetic fields [16], temperature, or electric fields. The above-mentioned fields were introduced into the high-energy ball milling process, and a variety of field-assisted high-energy ball milling technologies were developed. Thus, ball mill machinery can be organically combined with other physical energy, activating powders, accelerating powders’ structural thinning and promoting the mechanical alloying process [11,12]. For example, ultrasonic-assisted high-energy ball milling simultaneously carries out ultrasonic, mechanical and cavitation effects on powders, which can accelerate crushing, promote diffusion and achieve solid reaction.

Dielectric barrier discharge plasma-assisted milling (DBDPM), a new technology assisted by an external physical energy field [17], has begun to play a prominent role in the field of new material preparation [18]. DBDPM has a higher powder refining efficiency and more unique alloying mechanism than ordinary ball milling due to the temperature detonation effect, electron pulse effect and active group activation effect during plasma-assisted ball milling [19,20]. Therefore, in this study, DBDPM was employed for Al_2_O_3_ powder milling. The phase compositions, particle size distribution and microstructure of the Al_2_O_3_ powder after being milled various times were investigated. Furthermore, the properties and microstructure of alumina ceramics were prepared using the milled Al_2_O_3_ powders as starting materials. By analyzing the properties and microstructure of the fabricated Al_2_O_3_ ceramics, the activating mechanism of Al_2_O_3_ powder via DBDPM was explored.

## 2. Experimental

### 2.1. Raw Materials

The Al_2_O_3_ powder used in this study was provided by Jiangsu Jingxin New Materials Co., Ltd., Yangzhou, China. The chemical composition of the alumina powder is shown in Table 1.

### 2.2. Sample Preparation

The ball milling was carried out with a dielectric barrier discharge plasma-assisted ball mill (PBMS-mini, Xinwang Electronics Co., Ltd., Shenzhen, China). The ball mill was operated with a vibration frequency of 21.7 Hz, an amplitude of 10 mm, and a charge ratio of 10:1. As for the DBDPM, the discharge voltage was 5.4 kV, and the discharge frequency was 9.5 kHz. In order to evaluate the activation effect caused by discharge plasma, the plasma discharge power was turned off, and common vibratory milling (VM) without plasma assistance was carried out.

The Al_2_O_3_ powders milled for 1, 2, 3, 4 and 5 h were selected as the starting materials to fabricate alumina ceramics. Approximately 5 wt% polyvinyl alcohol (PVA solution with a concentration of 10 wt%) was added to the Al_2_O_3_ powders, and they were mixed for 30 min. The dried samples were vacuum-sealed and pre-formed under cold isostatic pressure of 150 MPa for 5 min. The pre-formed samples were fired at 1700 °C with a holding time of 3 h.

### 2.3. Characterization

The particle size of the Al_2_O_3_ powder after different times of ball milling was measured with a laser particle size analyzer (Mastersizer 2000, Malvern Instruments Co., Ltd., Illinois, UK). The surface area of the Al_2_O_3_ powder was determined using an automatic surface area and porosity analyzer (TriStar II 3020M, Micrometritics Instrument Corporation, Norcross, GA, USA).

As shown in Equations (1) and (2), using water as a medium, the bulk density (*ρ*_b_) and apparent porosity (*π*_a_) of the samples were measured based on Archimedes’ principle [4,5]. In the equations, *m*_1_, *m*_2_ and *m*_3_ are the dry, suspended and wet weights of the samples, respectively. In this study, more than ten measurements were made to measure the porosity and density of each sample.
(1)$$\rho_{b}=\frac{m_{1}}{m_{3}-m_{2}}$$
(2)$$\pi_{a}=\frac{m_{3}-m_{1}}{m_{3}-m_{2}}\times 100\%$$

The phase composition of each sample was analyzed using an X’PertPro MPD (PANalytical, Almelo, Holland) under the test conditions of Cu Ka ray, λ = 0.1542 nm, and with a scanning rate of 2 (°)/min. Microstructure observation and elemental analysis of the samples were performed with scanning electron microscopy (SEM, JSM-6610, JEOL, Tokyo, Japan) and using an attached energy dispersive spectrometer (EDS; QUANTAX, Bruker, Berlin, Germany).

The three-point prototype (ETM1050, China Guangdong Shenzhen Vance Testing Machine Co., Ltd., Shenzhen, China) was used to test the bending strength of each sample (10 × 10 × 60 mm^3^), with a span of 40 mm. Vickers hardness indentation was made on the polished sample surfaces using a Vickers hardness tester (HV-50A, Laizhou Huayin Testing Instrument Co., Ltd., Laizhou, China) equipped with a diamond pyramid indenter. A load of 5 kg and a loading time of 10 s were applied. Then, the cracks and indentations were observed using SEM, and Vickers hardness (*Hv*) and fracture toughness (*K_IC_*) were calculated according to Equations (3) and (4):(3)$$Hv=\frac{1.8544P}{102d^{2}}$$
(4)$$K_{IC}=0.028\times Hv^{0.5}\times E^{0.5}\times a^{2}\times c^{-1.5}$$

## 3. Results and Discussion

### 3.1. Alumina Powder

#### 3.1.1. Phase Compositions

Figure 1 shows the XRD patterns of the Al_2_O_3_ powder after different milling times. It can be seen that, with the extension of ball milling time, the broadening and dwarfing of Al_2_O_3_ diffraction peaks occur for both ball milling methods. Table 2 provides the full width at half maxima (FWHM) of the Al_2_O_3_ powder after different milling times. It can be seen that the FWHM increases with the increase in time, and the FWHM of Al_2_O_3_ after DBDPM is greater than that after VM. However, the crystal form of Al_2_O_3_ does not change and is still rhombohedral, as shown in Table 3. The broadening of diffraction peaks is mainly due to the linear broadening caused by stress release during grain refinement and the linear broadening caused by microscopic distortion.

#### 3.1.2. Particle Size

Table 4 presents a comparison between the particle sizes of the Al_2_O_3_ powder after different milling times. After the same ball milling time, it can be seen that the particle size (D_10_, D_50_ and D_90_) of the Al_2_O_3_ powder after DBDPM is smaller than that after VM. The particle size difference in D_50_ and D_90_ of the Al_2_O_3_ powder prepared with the two ball milling methods after the same milling time is about 1.2~1.5 µm and 0.4~0.7 µm, respectively, and the D_50_ particle size of the Al_2_O_3_ powder reaches 1.43 µm after 5 h of DBDPM. Figure 2 shows the specific surface area of the alumina powder after various milling times. The specific surface area of the alumina powder increases with the increasing milling time. Moreover, at the same ball milling time, the specific surface area of alumina powder after DBDPM is larger than that after VM. The specific surface area of the alumina powder after DBDPM and VM reached 2.2 m^2^/g and 1.76 m^2^/g, respectively, after 5 h of ball milling.

Compared with VM, DBDPM has both the mechanical force of a vibration ball mill and the thermal stress of plasma, and the two forces are superimposed to crush the Al_2_O_3_ particles. The electron temperature of plasma is very high, which can heat the micro-region of the powder instantaneously during ball milling. When leaving the plasma, the temperature of the powder drops sharply, which induces huge thermal stress, thus promoting powder crushing. In this way, the repeated process forms the powder refining mechanism of “melting-thermal explosion-quenching”.

As for VM, Al_2_O_3_ powder is strongly impacted by the milling balls, forming a large number of internal structural defects and increasing microscopic deformation. After repeated impact milling of the steel balls, the lattice relaxation of the Al_2_O_3_ powder occurs. With the increase in milling time, the particle size of the powder is decreased, and a large number of submicron particles are generated. In this case, the stress is released, leading to the broadening of X-ray diffraction peaks. However, when the ball milling goes on for a certain time, the grain size of Al_2_O_3_ becomes very small. At this time, the crushing capacity of single mechanical ball milling becomes smaller and smaller. The balance between the crushing effect and welding effect is achieved, and grain size and lattice distortion consequently tend to become stable. However, in DBDPM, the presence of plasma greatly increases the lattice distortion degree of Al_2_O_3_, resulting in a decreased particle size.

#### 3.1.3. Micromorphology Evolution of Al_2_O_3_ Powder in Ball Milling Process

The SEM image of the original unmilled Al_2_O_3_ powder is shown in Figure 3. It can be seen that the grains of the unmilled Al_2_O_3_ micropowder present an irregular shape, with an average particle size of around 7 µm.

Figure 4 provides the microstructure of Al_2_O_3_ powder after DBDPM for different amounts of time. The powder becomes finer with increasing ball milling time. The particle size of the alumina powder was approximately 3 µm after 3 h of ball milling. After 5 h of DBDPM, the average particle size decreased to around 1.5 µm, and the particle size was distributed mainly in the range of 500 nm~2 μm. However, some large bulks, with a size of about 5 µm, were observed (as shown in the circles in Figure 4e). This is because the powders are in an unstable state during the ball-milling process. Moreover, under the action of electrostatic attraction and van der Waals forces, the particles flock together spontaneously, thus reducing the enthalpy of the system and forming a soft agglomeration composed of secondary particles. Due to the action of high-energy electrons and active species, the agglomeration of DBDPM-milled powder is easier to occur than in VM-milled powder.

SEM images of the Al_2_O_3_ powder after different times of VM are given in Figure 5. The particle size of Al_2_O_3_ gradually decreased with the increase in ball milling time. After 1 h of ball milling, the Al_2_O_3_ had a wide primary particle size distribution (about 4~7μm), and the Al_2_O_3_ powder appeared as spherical aggregates. With the milling time going on to 3 h, the particle size of the Al_2_O_3_ powder was further decreased, but the morphology did not change significantly. When the ball milling time was further extended to 5 h, the morphology of the Al_2_O_3_ powder has changed obviously, and the particle size was further reduced. However, the particle size distribution was wider (approximately 500 nm~3 μm). Compared with the DBDPM-milled powder, the particle size of the VM-milled powder was much larger.

### 3.2. Alumina Ceramics

#### 3.2.1. Microstructure

SEM images of the polished surfaces of Al_2_O_3_ ceramics fabricated with various alumina powders are shown in Figure 6. It can be seen that the porosity of the alumina ceramics decreased with the increasing ball milling time. Moreover, the alumina ceramics fabricated from the DBDPM-milled powders possessed significantly lower porosity than that of ceramics fabricated from the VM-milled powders. The results indicate that DBDPM-milled powder shows better sintering activity than VM-milled powder.

Figure 7 presents the fracture surfaces of the Al_2_O_3_ ceramics. After the same ball milling time, the samples fabricated from DBDPM-milled powders are denser and the grain size is smaller. Moreover, compared with those fabricated from VM-milled powders, the samples fabricated from DBDPM-milled powders possess a more uniform grain distribution and a high length-diameter ratio of grains, which can not only increase flexural strength but also increase fracture toughness.

#### 3.2.2. Sintering Properties

The sintering properties of the Al_2_O_3_ ceramics fabricated from various alumina powders are shown in Figure 8. With an increasing milling time, the porosity of alumina ceramics decreases, while the bulk density increases. After the same ball milling time, the alumina ceramics prepared via DBDPM possess remarkably lower apparent porosity and significantly higher bulk density than that of the alumina ceramics fabricated from VM-milled powders. Because of the high-density and high-energy plasma bombardment in DBDPM, the method produces a great impact force and thermal effect on the surface of a powder. Moreover, it also leads to the local evaporation or melting of the material sputtering behavior. The vaporized components condense when they encounter other powder particles with lower temperatures, forming a large number of small primary particles on the surfaces of those particles. The highly active particles of the ionized body (ions, electrons, excited atoms and molecules, free radicals, etc.) are easy to adsorb with other substances and increase the activity of the material surface. The fresh surface and a large number of defects introduced by the ball mill further enhance the activity of the milled powder, making diffusion, phase transition and chemical reactions easier to occur. Therefore, in the sintering process, the alumina ceramics fabricated from DBDPM-milled powders were much denser than the alumina ceramics fabricated from VM-milled powders.

#### 3.2.3. Mechanical Properties

Figure 9 shows the flexural strength, Vickers hardness and fracture toughness of Al_2_O_3_ ceramics fabricated from various alumina powders. Compared with the VM-milled powders, the flexural strength, Vickers hardness and fracture toughness of alumina ceramic prepared from DBDPM-milled powders were slightly improved. After ball milling for 5 h, the alumina ceramic prepared from DBDPM-milled powders showed a 23 MPa higher flexural strength, a 0.7 GPa higher Vickers hardness and a 0.8 MPa m^1/2^ greater fracture toughness. This may be due to the smaller grain size of alumina ceramic (Figure 7) and the higher activity of alumina powder from DBDPM, thus promoting the density of sintering (Figure 8).

### 3.3. DBDPM Activation Mechanism

Comparing the grain size and lattice distortion of the Al_2_O_3_ powder after 5 h of ball milling between the two methods, it can be seen that the grain size of the Al_2_O_3_ powder after 5 h of ball milling with the two methods is roughly the same. However, the lattice distortion caused by DBDPM is much greater than that by VM. Therefore, one of the important reasons for the effective activation of Al_2_O_3_ powders via DBDPM is that the plasma results in greater lattice distortion, as shown in Figure 10.

In DBDPM, 0.1 MPa Ar gas is filled in the ball mill jar. When dielectric barrier discharge is implemented, a large number of charged particles in a high-energy state and optical radiation are generated in the dissociated Ar atmosphere [20]. These particles with high energy and high density collide and bombard the surface of the powder. When the excited particles with tens of electron volts bombard the powder surface, the incident particles and lattice atoms cause a series of collisions in the solid surface layer, which lead to the vibration or movement of lattice atoms. These energetic particles may be compounded by the surface of the powder or injected into the surface to transfer their kinetic energy to the lattice atoms. With an increase in DBDPM milling time, the density of incident particles on the Al_2_O_3_ powder increases, and a large number of lattice atoms in the crystal are in an unstable state. At the same time, the increase in the internal energy storage of the Al_2_O_3_ powder with dielectric properties leads to a sharp rise in powder temperature and even local melting. When the molten or semi-molten Al_2_O_3_ powder is frequently shear impacted by the steel ball at a low temperature, huge thermal stress is generated, which can be expressed as
(5)$$\sigma_{\max}=E\alpha \Delta T$$
where *E* is Young’s modulus, and *α* is the coefficient of thermal expansion.

Defects such as dislocation, vacancy and grain boundary are more likely to be formed inside the Al_2_O_3_ crystal, resulting in greater lattice distortion and a large number of fine microcrystalline particles. Therefore, the activation effect of DBDPM on Al_2_O_3_ powder is much greater than that of VM, which can effectively improve the sintering and mechanical properties of fabricated ceramics.

## 4. Conclusions

(1)Compared with VM, the grain size of Al_2_O_3_ powder obtained with DBDPM is smaller, and the lattice distortion is larger. The lattice distortion degree of DBDPM-milled Al_2_O_3_ powder shows an obvious increasing trend with the extension of assisted ball milling time.(2)In DBDPM, the thermal effect produced by plasma and the high-energy particle bombardment effect greatly promote the lattice distortion of Al_2_O_3_, which is an important reason for activating the Al_2_O_3_ reaction powder. Therefore, compared with the common vibratory milling technique, the fabricated alumina ceramics exhibited improved sintering and mechanical properties when using alumina powders produced via the DBDPM method as the starting materials.

## Figures and Tables

**Figure 1 materials-16-01128-f001:**
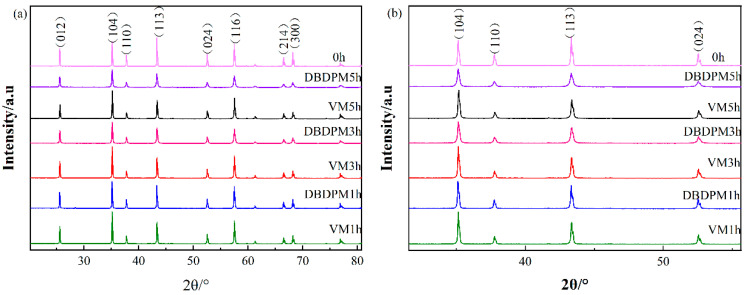
XRD patterns of Al_2_O_3_ powder after different milling times: (**a**) global drawing and (**b**) partial enlarged drawing.

**Figure 2 materials-16-01128-f002:**
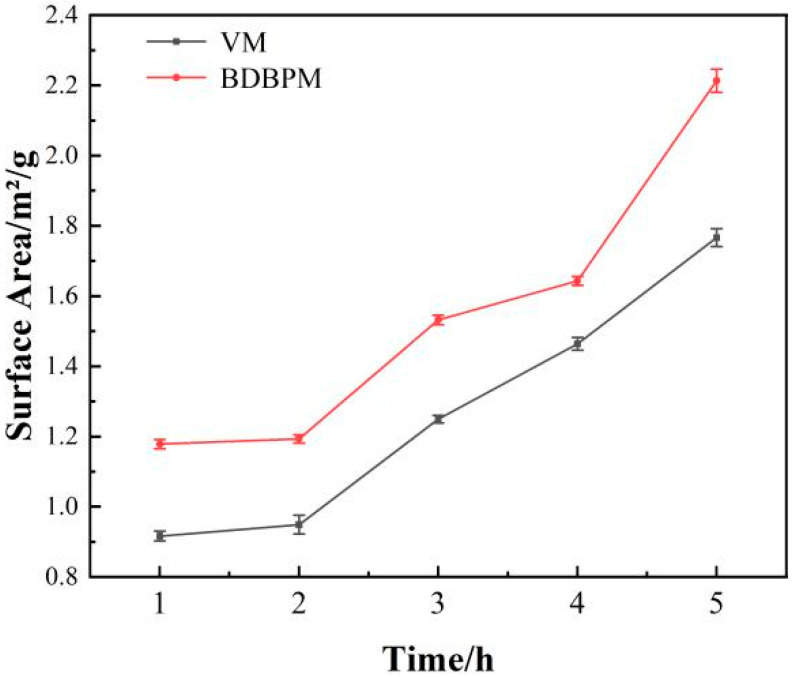
Surface area of Al_2_O_3_ powder after different milling times.

**Figure 3 materials-16-01128-f003:**
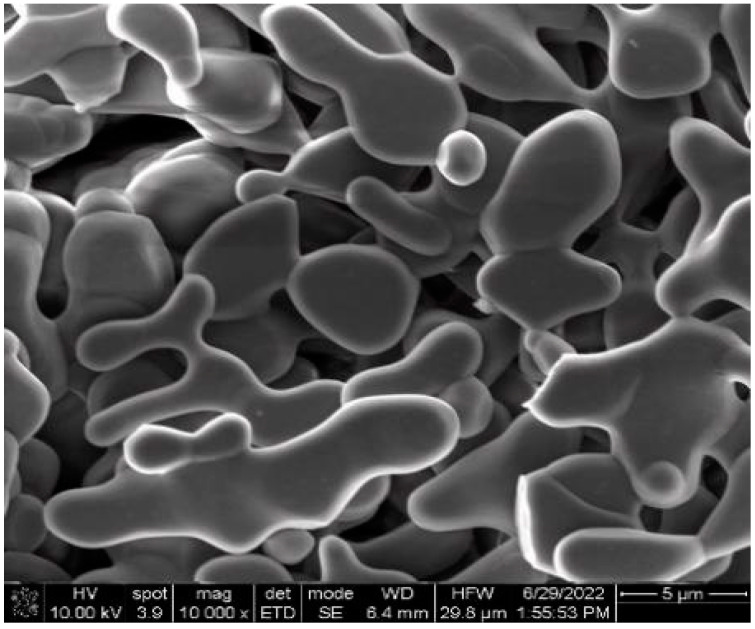
SEM image of the unmilled Al_2_O_3_ powder.

**Figure 4 materials-16-01128-f004:**
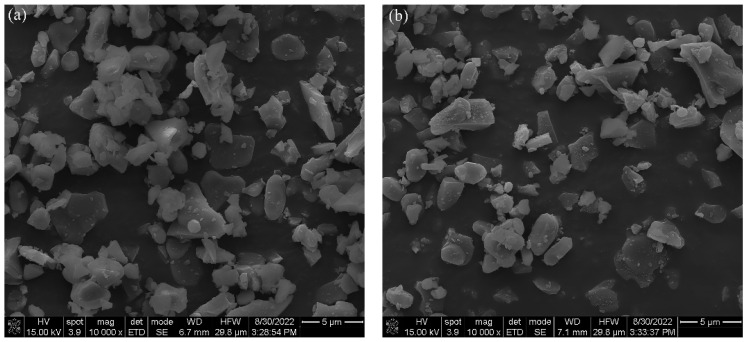
SEM images of Al_2_O_3_ powder after different times of DBDPM: (**a**) 1h; (**b**) 2 h; (**c**) 3 h; (**d**) 4 h; and (**e**) 5 h.

**Figure 5 materials-16-01128-f005:**
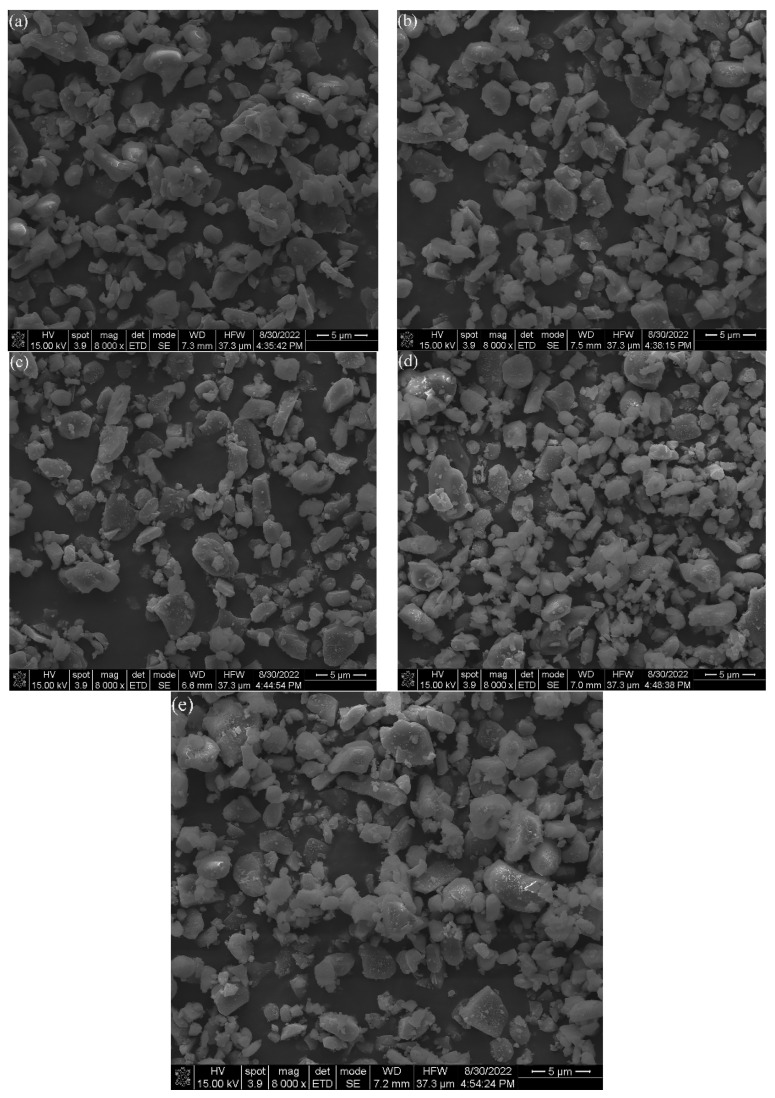
SEM images of Al_2_O_3_ powder after different times of VM: (**a**) 1 h; (**b**) 2 h; (**c**) 3 h; (**d**) 4 h; and (**e**) 5 h.

**Figure 6 materials-16-01128-f006:**
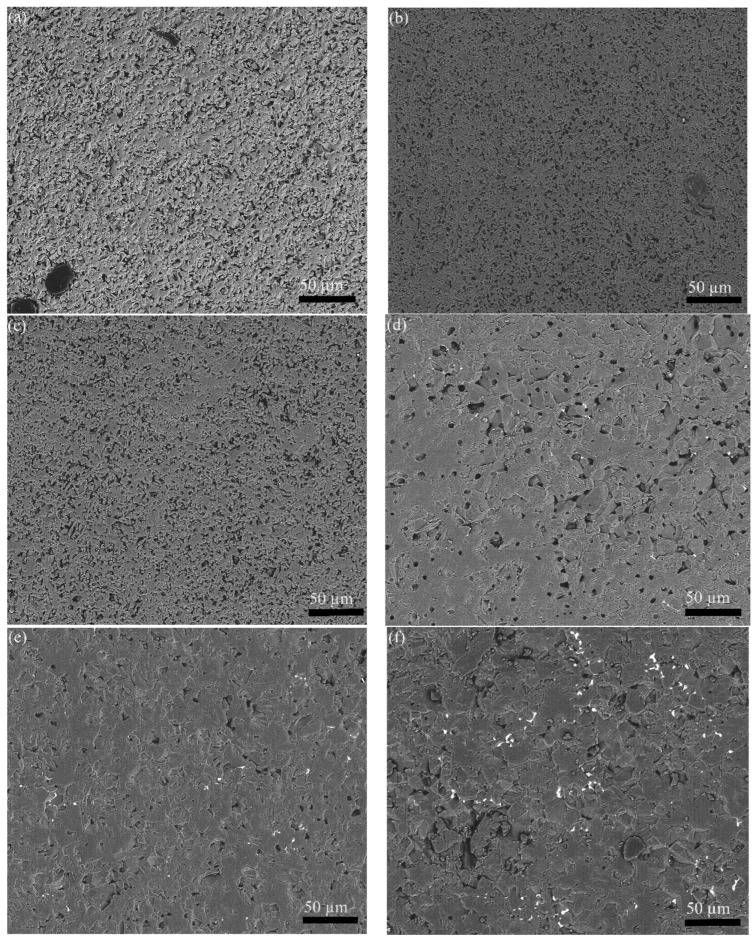
Polished surfaces of Al_2_O_3_ ceramics fabricated from various alumina powders: (**a**) VM 1 h; (**b**) VM 3 h; (**c**) VM 5 h; (**d**) DBDPM 1 h; (**e**) DBDPM 3 h; and (**f**) DBDPM 5 h.

**Figure 7 materials-16-01128-f007:**
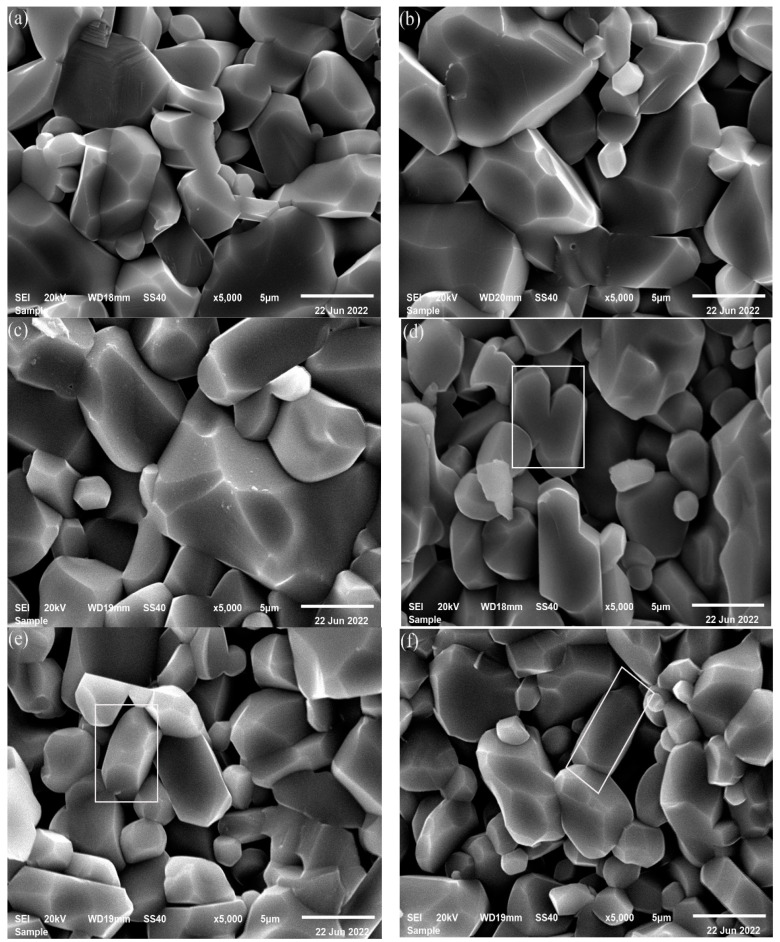
Fracture surfaces of Al_2_O_3_ ceramics fabricated from various alumina powders: (**a**) VM 1 h; (**b**) VM 3 h; (**c**) VM 5 h; (**d**) DBDPM 1 h; (**e**) DBDPM 3 h; and (**f**) DBDPM 5 h.

**Figure 8 materials-16-01128-f008:**
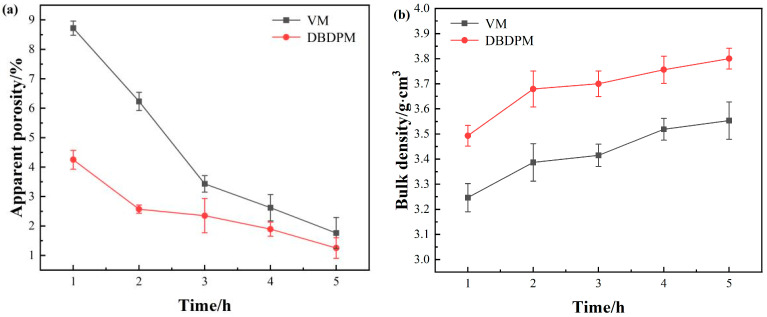
Sintering properties of Al_2_O_3_ ceramics fabricated from various alumina powders: (**a**) apparent porosity; (**b**) bulk density.

**Figure 9 materials-16-01128-f009:**
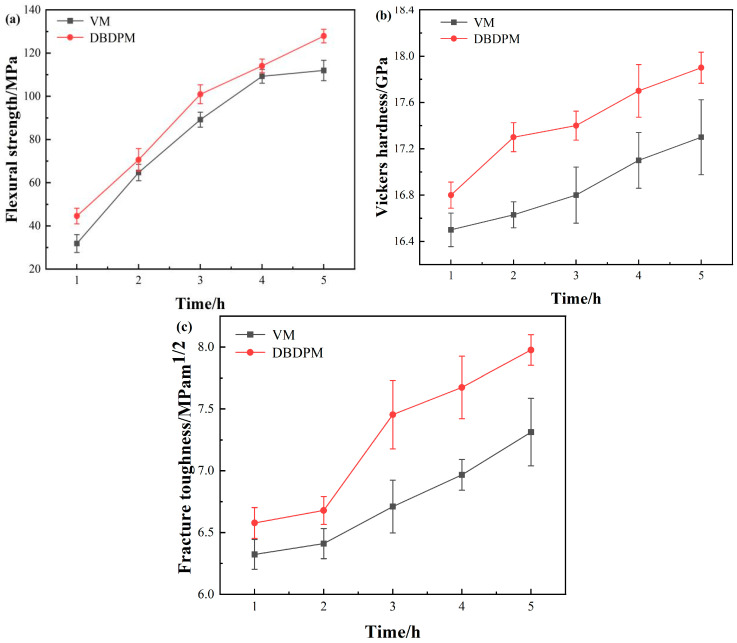
Mechanical properties of Al_2_O_3_ ceramics fabricated with various alumina powders: (**a**) flexural strength; (**b**) Vickers hardness; and (**c**) fracture toughness.

**Figure 10 materials-16-01128-f010:**
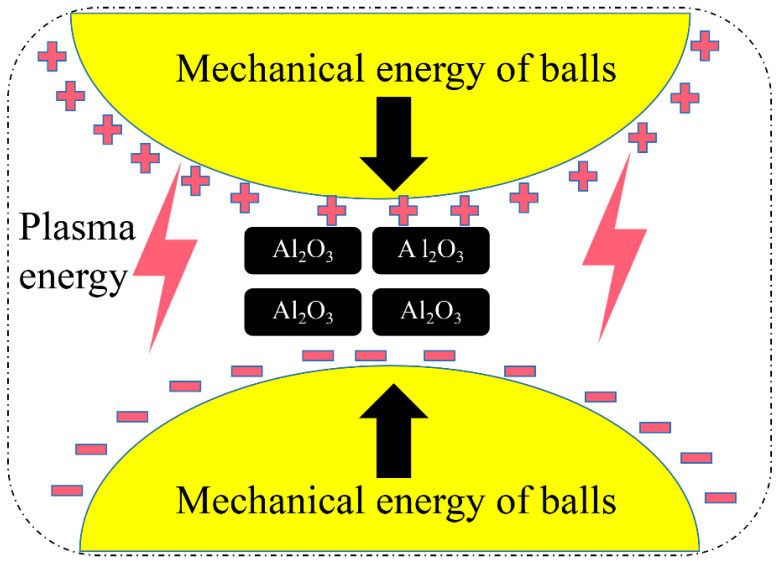
Schematic diagram of DBDPM activation mechanism on Al_2_O_3_ powder.

**Table 1 materials-16-01128-t001:** Chemical composition of the alumina powder (wt%).

Al_2_O_3_	SiO_2_	Fe_2_O_3_	CaO	MgO	K_2_O	Na_2_O	TiO_2_	ZrO_2_
99.59	0.052	0.033	0.013	0.087	0.011	0.16	0.015	0.001

**Table 2 materials-16-01128-t002:** FWHM of Al_2_O_3_ powder after different milling times.

	1 h	2 h	3 h
VM	0.15419	0.15863	0.16603
DBDP	0.15458	0.18687	0.20884

**Table 3 materials-16-01128-t003:** Crystal structure and lattice parameters of Al_2_O_3_.

Material	a (Å)	b (Å)	c (Å)	α = β (◦)	γ(◦)	Crystal System	Space Group
Al_2_O_3_	4.7581	4.7581	4.1815	90	120	Hexagonal	R-3C

**Table 4 materials-16-01128-t004:** Particle size of alumina powder after different milling times (µm).

	D_10_		D_50_		D_90_	
	VM	DBDPM	VM	DBDPM	VM	DBDPM
1 h	1.01	0.93	3.58	2.13	5.80	5.23
2 h	0.91	0.91	3.47	1.88	5.43	4.95
3 h	0.90	0.88	3.18	1.78	5.15	4.67
4 h	0.85	0.83	2.92	1.68	4.83	4.16
5 h	0.82	0.79	2.78	1.43	4.53	3.83

## Data Availability

Not applicable.

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
