# Peer review of "Improved Mechanical Properties of Alumina Ceramics Using Plasma-Assisted Milling Technique"

_materials, 2023, doi:10.3390/ma16031128_

Round 1

Reviewer 1 Report

Thank you very much for the nice written paper. The results are interesting but changes with regard to scientific issues has to be made. Please add standard deviations to the diagrams (grain size, bulk density, porosity, strength and so on) to show if there are significant differences.

Please add some explanations and references about the Voigt method for the calculation of the grain size and lattice distortion. You mentioned that you calculated the lattice distortion. Where are the results?

Please add information and references about the Archimedes Method - you did not mention the meaning of m1, m2 and m3.

As the grain size is one of the important results it would be necessary to show standard deviations and the term grain size is not clear. Typically, a d10, d50 and d90 or similar ones are used. This would give information about the distribution of the grain size. I would suggest adding a further method of measuring the grain size for example laser granulometry.

Author Response

Comment 1: Please add standard deviations to the diagrams (grain size, bulk density, porosity, strength and so on) to show if there are significant differences.

Response: Thanks for the reviewer’s suggestion, we have added standard deviations in the revised manuscript.

Comment 2: Please add some explanations and references about the Voigt method for the calculation of the grain size and lattice distortion. You mentioned that you calculated the lattice distortion. Where are the results?

Response: We are sorry for our negligence. At the beginning, we used Voigt method to calculate grain size, but we found that the limitation for crystallite size measurement is determined as max ~200 nm by XRD. Considering the limitation for crystallite size calculation and the actual particle size distribution of prepared alumina powders, we didn’t calculate the crystallite size using Voigt method. This sentence has already been deleted in the revised manuscript.

Comment 3: Please add information and references about the Archimedes Method - you did not mention the meaning of m1, m2 and m3.

Response: Thanks for the reviewer’s suggestion, we add explanations and references for m1, m2 and m3 in section 2.3.

Comment 4: As the grain size is one of the important results it would be necessary to show standard deviations and the term grain size is not clear. Typically, a d10, d50 and d90 or similar ones are used. This would give information about the distribution of the grain size. I would suggest adding a further method of measuring the grain size for example laser granulometry.

Response: Considering the reviewer’s suggestion, the d10, d50 and d90 of Al2O3 powder after different times of ball milling was measured by a laser particle size analyser (Mastersizer 2000, Malvern Instruments Co., Ltd., UK). And the results are provided in the revised manuscript.

Reviewer 2 Report

Dear Authority,

The manuscript entitled ‘Improved mechanical properties of alumina ceramics by plasma-assisted milling technique’ offer a comparison between dielectric barrier dis-charge plasma-assisted milling (DBDPM) and  common vibratory milling technique in terms of powder morphology and mechanical properties of bulk samples. Authors suggests that DBDPM technique provides smaller powder particles size and improvement on the hardness (Hv) and fracture toughness of bulk samples. Even though there is valuable information about the contribution of dielectric barrier dis-charge plasma-assisted milling (DBDPM), there are few missing points in experimental procedure which restricts to make reliable assessment about endproduct. The manuscript needs to be revisited by considering following comments;

1- At page 4, it is stated as ‘The pre-formed samples were fired at 1600 ℃ and 1700 ℃ respectively, and the holding time was 3h for both’. This sentences is not clear. Please provide the sintering temperature for each condition. If you choose different temperature value for each condition, you cannot make assessment about endproduct and whole manuscript becomes meaningless. For comparison of both condition, same production condition needs to be selected.

2- In section 3.1.2, comparison of particle size of Al2O3 powder after different milling times is presented. However, it is not stated how to measure particle size for both condition in manuscript body.

3- In xrd data, FWHM for each condition is reduced with increasing milling time. So, you can make crystallite size calculation by Scherrer equation. It is important note that the limitation for crystallite size measurement is determined as max ~200 nm by XRD. Therefore, the micron level for grain or particle size distribution in your system hinders to make assessment about crystallite size. However, changes in FWHM gives an idea about the degree of crystallinity. Please pay attention regarding crystallite size calculation.

3- SEM illustration for both condition, I cannot detect major differences at particle size distribution comparing 5h milling conditions for both milling technique. For clear comparison, I recommend to make measurement of the average particle size (APS) and specific surface area (SSA) of milled powders by Mastersizer 2000 particle size analyzer instrument.

4- There is no error bars for figure 2, 8 and 9. Please revisit the graphs

After major modification, the paper could be considered for publication in Materials.

Best wishes,

Author Response

Comment 1: At page 4, it is stated as‘The pre-formed samples were fired at 1600 ℃ and 1700 ℃ respectively, and the holding time was 3h for both’. This sentences is not clear. Please provide the sintering temperature for each condition. If you choose different temperature value for each condition, you cannot make assessment about endproduct and whole manuscript becomes meaningless. For comparison of both condition, same production condition needs to be selected.

Response: Thanks for the reviewer’s reminder. In fact, we have compared the effect of sintering temperature on the properties of fabricated alumina ceramics in our study. However, in this paper, all the samples were sintered only at 1700℃. So, that is a clerical error, we have corrected this sentence in the revised manuscript.

Comment 2: In section 3.1.2, comparison of particle size of Al2O3 powder after different milling times is presented. However, it is not stated how to measure particle size for both condition in manuscript body.

Response: The particle size of Al2O3 powder after different times of ball milling was measured by a laser particle size analyser (Mastersizer 2000, Malvern Instruments Co., Ltd., UK). We have already added this information in the revised manuscript. 

Comment 3: In xrd data, FWHM for each condition is reduced with increasing milling time. So, you can make crystallite size calculation by Scherrer equation. It is important note that the limitation for crystallite size measurement is determined as max ~200 nm by XRD. Therefore, the micron level for grain or particle size distribution in your system hinders to make assessment about crystallite size. However, changes in FWHM gives an idea about the degree of crystallinity. Please pay attention regarding crystallite size calculation.

Response: Thanks very much for the reviewer’s valuable comment. Yes, considering the limitation for crystallite size calculation by Scherrer equation and the actual particle size distribution of prepared alumina powders, we didn’t calculate the crystallite size using Scherrer equation. We would thank you again for your kind suggestion.

Comment 4: SEM illustration for both condition, I cannot detect major differences at particle size distribution comparing 5h milling conditions for both milling technique. For clear comparison, I recommend to make measurement of the average particle size (APS) and specific surface area (SSA) of milled powders by Mastersizer 2000 particle size analyzer instrument.

Response: Considering the reviewer’s suggestion, the specific surface area and d10, d50 and d90 of Al2O3 powder after different times of ball milling was provided in the revised manuscript.

Comment 5: There is no error bars for figure 2, 8 and 9. Please revisit the graphs.

Response: Thanks for the reviewer’s suggestion, we have added standard deviations in the revised manuscript.

Round 2

Reviewer 1 Report

Authors are presenting a comprehensive study comparing two types of milling technique. The authors show various effects on the different milling techniques. The describion of the methods can be improved by giving more parameter for example at the Malvern Measurement. The results are clearly presented. 

The demonstrated work would have a good impact for the scientific and industrial community.

Reviewer 2 Report

The authors have been made the revision accoording to my comments. So, the paper is now suitable to publish in Materials.